# ENABLING BINARY NEURAL NETWORK TRAINING ON THE EDGE

## ABSTRACT

The ever-growing computational demands of increasingly complex machine learning models frequently necessitate the use of powerful cloud-based infrastructure for their training. Binary neural networks are known to be promising candidates for on-device inference due to their extreme compute and memory savings over higher-precision alternatives. In this paper, we demonstrate that they are also strongly robust to gradient quantization, thereby making the training of modern models on the edge a practical reality. We introduce a low-cost binary neural network training strategy exhibiting sizable memory footprint reductions and energy savings vs Courbariaux & Bengio's standard approach. Against the latter, we see coincident memory requirement and energy consumption drops of 2–6×, while reaching similar test accuracy in comparable time, across a range of small-scale models trained to classify popular datasets. We also showcase ImageNet training of ResNetE-18, achieving a 3.12× memory reduction over the aforementioned standard. Such savings will allow for unnecessary cloud offloading to be avoided, reducing latency and increasing energy efficiency while also safeguarding privacy.

## 1 INTRODUCTION

Although binary neural networks (BNNs) feature weights and activations with just single-bit precision, many models are able to reach accuracy indistinguishable from that of their higher-precision counterparts (Courbariaux & Bengio, 2016; Wang et al., 2019b). Since BNNs are functionally complete, their limited precision does not impose an upper bound on achievable accuracy (Constantinides, 2019). BNNs represent the ideal class of neural networks for edge inference, particularly for custom hardware implementation, due to their use of XNOR for multiplication: a fast and cheap operation to perform. Their use of compact weights also suits systems with limited memory and increases opportunities for caching, providing further potential performance boosts. FINN, the seminal BNN implementation for field-programmable gate arrays (FPGAs), reached the highest CIFAR-10 and SVHN classification rates to date at the time of its publication (Umuroglu et al., 2017).

Despite featuring binary forward propagation, existing BNN training approaches perform backward propagation using high-precision floating-point data types—typically `float32`—often making training infeasible on resource-constrained devices. The high-precision activations used between forward and backward propagation commonly constitute the largest proportion of the total memory footprint of a training run (Sohoni et al., 2019; Cai et al., 2020). Additionally, backward propagation with high-precision gradients is costly, challenging the energy limitations of edge platforms.

An understanding of standard BNN training algorithms led us to ask two questions: why are high-precision weight gradients used when we are only concerned with weights' *signs*, and why are high-precision activations used when the computation of weight gradients only requires *binary* activations as input? In this paper, we present a low-memory, low-energy BNN training scheme based on this intuition featuring (i) the use of binary, power-of-two and 16-bit floating-point data types, and (ii) batch normalization modifications enabling the buffering of binary activations.

By increasing the viability of learning on the edge, this work will reduce the domain mismatch between training and inference—particularly in conjunction with federated learning (McMahan et al., 2017; Bonawitz et al., 2019)—and ensure privacy for sensitive applications (Agarwal et al., 2018). Via the aggressive energy and memory footprint reductions they facilitate, our proposals will enable

Table 1: Comparison of applied approximations vs related low-cost neural network training works.

| | Weights | Weight gradients | Activations | Activation gradients | Batch normalization |
|---|---|---|---|---|---|
| Zhou et al. (2016) | `int6`[1] | `int6` | `int6` | `int6` | ✘ |
| Gruslys et al. (2016) | ✘ | ✘ | Recomputed[2] | ✘ | ✘ |
| Ginsburg et al. (2017) | `float16` | `float16` | `float16` | `float16` | ✘ |
| Graham (2017) | ✘ | ✘ | `int` | ✘ | ✘ |
| Bernstein et al. (2018) | ✘ | `bool` | ✘ | ✘ | ✘ |
| Wu et al. (2018b) | ✘ | ✘ | ✘ | ✘ | $l_1$ |
| **This work** | `bool` | `bool` | `bool` | `po2`[3] | BNN-specific $l_1$ |

[1] Arbitrary precision was supported, but significant accuracy degradation was observed below 6 bits.
[2] Activations were not retained between forward and backward propagation in order to save memory.
[3] Power-of-two format comprising sign bit and exponent.

networks to be trained without the network access reliance, latency and energy overheads or data divulgence inherent to cloud offloading. To this end, we make the following novel contributions.

- We conduct the first variable representation and lifetime analysis of the standard BNN training process, informing the application of beneficial approximations. In particular, we
  - binarize weight gradients owing to the lack of importance of their magnitudes,
  - modify the forward and backward batch normalization operations such that we remove the need to buffer high-precision activations and
  - determine and apply appropriate additional quantization schemes—power-of-two activation gradients and reduced-precision floating-point data—taken from the literature.
- Against Courbariaux & Bengio (2016)'s approach, we demonstrate the preservation of test accuracy and convergence rate when training BNNs to classify MNIST, CIFAR-10, SVHN and ImageNet while lowering memory and energy needs by up to $5.67\times$ and $4.53\times$.
- We provide an open-source release of our training software, along with our memory and energy estimation tools, to the community[1].

## 2   RELATED WORK

The authors of all published works on BNN inference acceleration to date made use of high-precision floating-point data types during training (Courbariaux et al., 2015; Courbariaux & Bengio, 2016; Lin et al., 2017; Ghasemzadeh et al., 2018; Liu et al., 2018; Wang et al., 2019a; 2020; Umuroglu et al., 2020; He et al., 2020; Liu et al., 2020). There is precedent, however, for the use of quantization when training non-binary networks, as we show in Table 1 via side-by-side comparison of the approximation approaches taken in those works along with that proposed herein.

The effects of quantizing the gradients of networks with high-precision data, either fixed or floating point, have been studied extensively. Zhou et al. (2016) and Wu et al. (2018a) trained networks with fixed-point weights and activations using fixed-point gradients, reporting no accuracy loss for AlexNet classifying ImageNet with gradients wider than five bits. Wen et al. (2017) and Bernstein et al. (2018) focused solely on aggressive weight gradient quantization, aiming to reduce communication costs for distributed learning. Weight gradients were losslessly quantized into ternary and binary formats, respectively, with forward propagation and activation gradients kept at high precision. In this work, we make the novel observations that activation gradient dynamic range is more important than precision, and that BNNs are more robust to approximation than higher-precision networks. We thus propose a data representation scheme more aggressive than all of the aforementioned works combined, delivering large memory and energy savings with near-lossless performance.

Gradient checkpointing—the recomputation of activations during backward propagation—has been proposed as a method to reduce the memory consumption of training (Chen et al., 2016; Gruslys

---
[1]Source supplied in `.zip` for review.

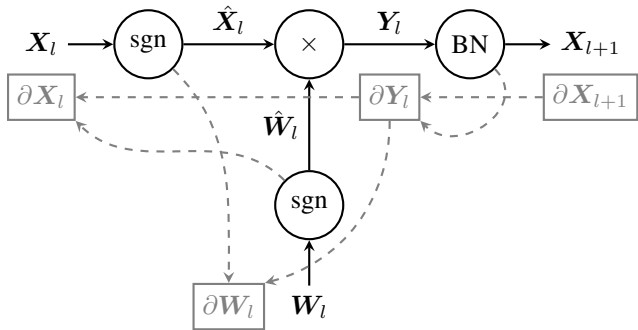

Figure 1: Standard BNN training graph for fully connected layer $l$. "sgn," "$\times$" and "BN" are sign, matrix multiplication and batch normalization operations. Forward propagation dependencies are shown in black; those for backward passes are gray.

et al., 2016). Such methods introduce additional forward passes, however, and so increase each run's duration and energy cost. Graham (2017) and Chakrabarti & Moseley (2019) saved memory during training by buffering activations in low-precision formats, achieving comparable accuracy to all-`float32` baselines. Wu et al. (2018b) and Hoffer et al. (2018) reported reduced computational costs via $l_1$ batch normalization. Finally, Helwegen et al. (2019) asserted that the use of both trainable weights and momenta is superfluous in BNN optimizers, proposing a weightless BNN-specific optimizer, Bop, able to reach the same level of accuracy as Adam. We took inspiration from these works in locating sources of redundancy present in standard BNN training schemes, and propose BNN-specific modifications to $l_1$ batch normalization allowing for activation quantization all the way down to binary, thus saving both memory and energy without inducing latency increases.

## 3 STANDARD TRAINING FLOW

For simplicity, we assume the use of a multi-layer perceptron (MLP), although the presence of convolutional layers would not change any of the principles that follow. Let $\boldsymbol{W}_l$ and $\boldsymbol{X}_l$ denote matrices of weights and activations, respectively, in the network's $l^{\text{th}}$ layer, with $\partial \boldsymbol{W}_l$ and $\partial \boldsymbol{X}_l$ being their gradients. For $\boldsymbol{W}_l$, rows and columns span input and output channels, respectively, while for $\boldsymbol{X}_l$ they represent feature maps and channels. Henceforth, we use decoration to indicate low-precision data representation, with $\hat{\bullet}$ and $\tilde{\bullet}$ respectively denoting binary and power-of-two encoding.

Figure 1 shows the training graph of a fully connected binary layer. A detailed description of the standard BNN training procedure introduced by Courbariaux & Bengio (2016) for each batch of $B$ training samples, which we henceforth refer to as as a *step*, is provided in Algorithm 1. Therein, "$\odot$" signifies element-wise multiplication. For brevity, we omit some of the intricacies of the baseline implementation—lack of first-layer quantization, use of a final softmax layer and the inclusion of weight gradient cancellation (Courbariaux & Bengio, 2016)—as these standard BNN practices are not impacted by our work. We initialize weights as outlined by Glorot & Bengio (2010).

Many authors have found that BNNs require batch normalization in order to avoid gradient explosion (Alizadeh et al., 2018; Sari et al., 2019; Qin et al., 2020), and our early experiments confirmed this to indeed be the case. We thus apply it as standard. Matrix products $\boldsymbol{Y}_l$ are channel-wise batch-normalized across each layer's $M_l$ output channels (Algorithm 1 lines 5–6) to form the subsequent layer's inputs, $\boldsymbol{X}_{l+1}$. $\boldsymbol{\beta}$ constitutes the batch normalization biases. Layer-wise moving means $\mu(\boldsymbol{y}_l)$ and standard deviations $\sigma(\boldsymbol{y}_l)$ are retained for use during backward propagation and inference. We forgo trainable scaling factors, commonly denoted $\boldsymbol{\gamma}$; these are of irrelevance to BNNs since their activations are binarized prior to use during forward propagation (line 2).

## 4 VARIABLE ANALYSIS

In order to quantify the potential gains from approximation, we conducted a variable representation and lifetime analysis of Algorithm 1 following the approach taken by Sohoni et al. (2019). Table 2

**Algorithm 1** Standard BNN training step.

1: **for** $l \leftarrow \{1, \cdots, L-1\}$ **do** $\qquad \triangleright$ Forward
2: $\qquad \hat{\boldsymbol{X}}_l \leftarrow \text{sgn}(\boldsymbol{X}_l)$
3: $\qquad \hat{\boldsymbol{W}}_l \leftarrow \text{sgn}(\boldsymbol{W}_l)$
4: $\qquad \boldsymbol{Y}_l \leftarrow \hat{\boldsymbol{X}}_l \hat{\boldsymbol{W}}_l$
5: $\qquad$ **for** $m \leftarrow \{1, \cdots, M_l\}$ **do**
6: $\qquad\qquad \boldsymbol{x}_{l+1}^{(m)} \leftarrow \dfrac{\boldsymbol{y}_l^{(m)} - \mu\big(\boldsymbol{y}_l^{(m)}\big)}{\sigma\big(\boldsymbol{y}_l^{(m)}\big)} + \beta_l^{(m)}$
7: **for** $l \leftarrow \{L-1, \cdots, 1\}$ **do** $\qquad \triangleright$ Backward
8: $\qquad$ **for** $m \leftarrow \{1, \cdots, M_l\}$ **do**
9: $\qquad\qquad \boldsymbol{v} \leftarrow \dfrac{1}{\sigma\big(\boldsymbol{y}_l^{(m)}\big)} \partial \boldsymbol{x}_{l+1}^{(m)}$
10: $\qquad\qquad \partial \boldsymbol{y}_l^{(m)} \leftarrow \boldsymbol{v} - \mu(\boldsymbol{v}) - $
$\qquad\qquad\qquad\qquad \mu\big(\boldsymbol{v} \odot \boldsymbol{x}_{l+1}^{(m)}\big)\boldsymbol{x}_{l+1}^{(m)}$
11: $\qquad\qquad \partial \beta_l^{(m)} \leftarrow \sum \partial \boldsymbol{x}_{l+1}^{(m)}$
12: $\qquad \partial \boldsymbol{X}_l \leftarrow \partial \boldsymbol{Y}_l \hat{\boldsymbol{W}}_l^{\mathrm{T}}$
13: $\qquad \partial \boldsymbol{W}_l \leftarrow \hat{\boldsymbol{X}}_l^{\mathrm{T}} \partial \boldsymbol{Y}_l$
14: **for** $l \leftarrow \{1, \cdots, L-1\}$ **do** $\qquad \triangleright$ Update
15: $\qquad \boldsymbol{W}_l \leftarrow \text{Optimize}(\boldsymbol{W}_l, \partial \boldsymbol{W}_l, \eta)$
16: $\qquad \boldsymbol{\beta}_l \leftarrow \text{Optimize}(\boldsymbol{\beta}_l, \partial \boldsymbol{\beta}_l, \eta)$
17: $\eta \leftarrow \text{LearningRateSchedule}(\eta)$

**Algorithm 2** Proposed BNN training step.

1: **for** $l \leftarrow \{1, \cdots, L-1\}$ **do** $\qquad \triangleright$ Forward
2: $\qquad \hat{\boldsymbol{X}}_l \leftarrow \text{sgn}(\boldsymbol{X}_l)$
3: $\qquad \hat{\boldsymbol{W}}_l \leftarrow \text{sgn}(\boldsymbol{W}_l)$
4: $\qquad \boldsymbol{Y}_l \leftarrow \hat{\boldsymbol{X}}_l \hat{\boldsymbol{W}}_l$
5: $\qquad$ **for** $m \leftarrow \{1, \cdots, M_l\}$ **do**
6: $\qquad\qquad \boldsymbol{x}_{l+1}^{(m)} \leftarrow \dfrac{\boldsymbol{y}_l^{(m)} - \mu\big(\boldsymbol{y}_l^{(m)}\big)}{\big\| \boldsymbol{y}_l^{(m)} - \mu\big(\boldsymbol{y}_l^{(m)}\big) \big\|_1 / B} + \beta_l^{(m)}$
7: **for** $l \leftarrow \{L-1, \cdots, 1\}$ **do** $\qquad \triangleright$ Backward
8: $\qquad$ **for** $m \leftarrow \{1, \cdots, M_l\}$ **do**
9: $\qquad\qquad \boldsymbol{v} \leftarrow \dfrac{1}{\big\| \boldsymbol{y}_l^{(m)} - \mu\big(\boldsymbol{y}_l^{(m)}\big) \big\|_1 / B} \partial \boldsymbol{x}_{l+1}^{(m)}$
10: $\qquad\qquad \partial \boldsymbol{y}_l^{(m)} \leftarrow \boldsymbol{v} - \mu(\boldsymbol{v}) - $
$\qquad\qquad\qquad \mu\big(\boldsymbol{v} \odot \hat{\boldsymbol{x}}_{l+1}^{(m)} \big\| \boldsymbol{x}_{l+1}^{(m)} \big\|_1 / B\big)\hat{\boldsymbol{x}}_{l+1}^{(m)}$
11: $\qquad\qquad \partial \beta_l^{(m)} \leftarrow \sum \partial \boldsymbol{x}_{l+1}^{(m)}$
12: $\qquad \partial \tilde{\boldsymbol{Y}}_l \leftarrow \text{po2}(\partial \boldsymbol{Y}_l)$
13: $\qquad \partial \boldsymbol{X}_l \leftarrow \partial \tilde{\boldsymbol{Y}}_l \hat{\boldsymbol{W}}_l^{\mathrm{T}}$
14: $\qquad \partial \boldsymbol{W}_l \leftarrow \hat{\boldsymbol{X}}_l^{\mathrm{T}} \partial \tilde{\boldsymbol{Y}}_l$
15: $\qquad \partial \hat{\boldsymbol{W}}_l \leftarrow \text{sgn}(\partial \boldsymbol{W}_l)$
16: **for** $l \leftarrow \{1, \cdots, L-1\}$ **do** $\qquad \triangleright$ Update
17: $\qquad \boldsymbol{W}_l \leftarrow \text{Optimize}\big(\boldsymbol{W}_l, \partial \hat{\boldsymbol{W}}_l / \sqrt{M_{l-1}}, \eta\big)$
18: $\qquad \boldsymbol{\beta}_l \leftarrow \text{Optimize}(\boldsymbol{\beta}_l, \partial \boldsymbol{\beta}_l, \eta)$
19: $\eta \leftarrow \text{LearningRateSchedule}(\eta)$

lists the properties of all variables in Algorithm 1, with each variable's contributions to the total footprint shown for a representative example. Variables are divided into two classes: those that must remain in memory between computational phases (forward propagation, backward propagation and weight update), and those that need not. This is of pertinence since, for those in the latter category, only the largest layer's contribution counts towards the total memory occupancy. For example, $\partial \boldsymbol{X}_l$ is read during the backward propagation of layer $l-1$ only, thus $\partial \boldsymbol{X}_{l-1}$ can safely overwrite $\partial \boldsymbol{X}_l$ for efficiency. Additionally, $\boldsymbol{Y}$ and $\partial \boldsymbol{X}$ are shown together since they are equally sized and only need to reside in memory during the forward and backward pass for each layer, respectively.

## 5 LOW-COST BNN TRAINING

As shown in Table 2, all variables within the standard BNN training flow use `float32` representation. In the subsections that follow, we detail the application of aggressive approximation specifically tailored to BNN training. Further to this, and in line with the observation by many authors that `float16` can be used for ImageNet training without inducing accuracy loss (Ginsburg et al., 2017; Wang et al., 2018; Micikevicius et al., 2018), we also switch all remaining variables to this format. Our final training procedure is captured in Algorithm 2, with modifications from Algorithm 1 in red and the corresponding data representations used shown in bold in Table 2. We provide both theoretical evidence and training curves for all of our experiments in Appendix A to show that our proposed approximations do not cause material degradation to convergence rates.

### 5.1 GRADIENT QUANTIZATION

**Binarized weight gradients.** Intuitively, BNNs should be particularly robust to weight gradient binarization since their weights only constitute signs. On line 15 of Algorithm 2, therefore, we quantize and store weight gradients in binary format, $\partial \hat{\boldsymbol{W}}$, for use during weight update. During the latter, we attenuate the gradients by $\sqrt{N_l}$, where $N_l$ is layer $l$'s fan-in, to reduce the learning rate

Table 2: Memory-related properties of variables used during training. To obtain the exemplary quantities of total memory given, BinaryNet was trained for CIFAR-10 classification with Adam.

| Variable | Per-layer lifetime[1] | Standard training | | | Proposed training | | |
|---|---|---|---|---|---|---|---|
| | | Data type | Size (MiB) | % | Data type | Size (MiB) | Saving ($\times$) |
| $\boldsymbol{X}$ | ✗ | `float32` | 111.33 | 26.18 | `bool` | **3.48** | **32.00** |
| $\partial\boldsymbol{X}, \boldsymbol{Y}^2$ | ✔ | `float32` | 50.00 | 11.76 | `float16` | **25.00** | **2.00** |
| $\mu(\boldsymbol{y}_l)$ | ✗ | `float32` | 0.01 | 0.00 | `float16` | **0.01** | **2.00** |
| $\sigma(\boldsymbol{y}_l)$ | ✗ | `float32` | 0.01 | 0.00 | `float16` | **0.01** | **2.00** |
| $\partial\boldsymbol{Y}$ | ✔ | `float32` | 50.00 | 11.76 | `po2_5`[3] | **7.81** | **6.40** |
| $\boldsymbol{W}$ | ✗ | `float32` | 53.49 | 12.58 | `float16` | **26.74** | **2.00** |
| $\partial\boldsymbol{W}$ | ✗ | `float32` | 53.49 | 12.58 | `bool` | **1.67** | **32.00** |
| $\boldsymbol{\beta}$ | ✗ | `float32` | 0.01 | 0.00 | `float16` | **0.01** | **2.00** |
| $\partial\boldsymbol{\beta}$ | ✗ | `float32` | 0.01 | 0.00 | `float16` | **0.01** | **2.00** |
| Momenta | ✗ | `float32` | 106.98 | 25.15 | `float16` | **53.49** | **2.00** |
| Total | | | 425.33 | 100.00 | | **118.23** | **3.60** |

[1] Variables that need not be retained between forward, backward or update phases of Algorithms 1 and 2.
[2] $\partial\boldsymbol{X}$ and $\boldsymbol{Y}$ can share memory since they are equally sized and have non-overlapping lifetime.
[3] 5-bit power-of-two format with 4-bit exponent.

and prevent premature weight clipping as advised by Sari et al. (2019). Since fully connected layers are used as an example in Algorithm 2, $N_l = M_{l-1}$ in this instance.

Table 2 shows that, with binarization, the portion of our exemplary training run's memory consumption attributable to weight gradients dropped from 53.49 to just 1.67 MiB, leaving the scarce resources available for more quantization-sensitive variables such as $\boldsymbol{W}$ and momenta. Energy consumption will also decrease due to the associated reduction in memory traffic.

**Power-of-two activation gradients.** The tolerance of BNN training to weight gradient binarization further suggests that activation gradients can be aggressively approximated without causing significant accuracy loss. Unlike previous proposals, in which activation gradients were quantized into fixed- or block floating-point formats (Zhou et al., 2016; Wu et al., 2018a), we hypothesize that power-of-two representation is more suitable due to their typically high inter-channel variance.

We define power-of-two quantization into $k$-bit "`po2_k`" format as $\mathrm{po2}_k(\bullet) = \mathrm{sgn}(\bullet) \odot 2^{\exp(\bullet)-b}$, comprising a sign bit and $k-1$-bit exponent $\exp(\bullet) = \max\left(-2^{k-2}, [\log_2(\bullet) + b]\right)$ with bias $b = 2^{k-2} - 1 - [\log_2(\|\bullet\|_\infty)]$. Square brackets signify rounding to the nearest integer. With $b$, we scale $\exp(\bullet)$ layer-wise to make efficient use of its dynamic range. This is applied to quantize matrix product gradients $\partial\boldsymbol{Y}_l$ on line 12 of Algorithm 2. We chose to use $k = 5$ as standard, generally finding this value to result in high compression while inducing little loss in accuracy. While we elected not to similarly approximate $\partial\boldsymbol{X}$ due to its use in the computation of quantization-sensitive $\boldsymbol{\beta}$, our use of of $\partial\tilde{\boldsymbol{Y}} = \mathrm{po2}(\partial\boldsymbol{Y})$ nevertheless leads to sizeable reductions in total memory footprint. Our use of $\partial\tilde{\boldsymbol{Y}}$ further allows us to reduce the energy consumption associated with lines 13–14 in Algorithm 2, for both of which we now have one binary and one power-of-two operand. Assuming that the target training platform has native support for only 32-bit fixed- and floating-point arithmetic, these matrix multiplications can be computed by (i) converting powers-of-two into `int32s` via shifts, (ii) performing sign-flips and (iii) accumulating the `int32` outputs. This consumes far less energy than the standard training method's all-`float32` equivalent.

## 5.2 BATCH NORMALIZATION APPROXIMATION

Analysis of the backward pass of Algorithm 1 reveals conflicting requirements for the precision of $\boldsymbol{X}$. When computing weight gradients $\partial\boldsymbol{W}$ (line 13), only binary activations $\hat{\boldsymbol{X}}$ are needed. For the batch normalization training (lines 8–11), however, high-precision $\boldsymbol{X}$ is used. As was shown in Table 2, the storage of $\boldsymbol{X}$ between forward and backward propagation constitutes the single largest portion of the algorithm's total memory. If we are able to use $\hat{\boldsymbol{X}}$ in place of $\boldsymbol{X}$ for these operations,

there will be no need to retain the high-precision activations, significantly reducing memory footprint as a result. We achieve this goal via the following two steps.

**Step 1: $l_1$ normalization.** Standard batch normalization sees channel-wise $l_2$ normalization performed on each layer's centralized activations. Since batch normalization is immediately followed by binarization in BNNs, however, we argue that less-costly $l_1$ normalization is good enough in this circumstance. Replacement of batch normalization's backward propagation operation with our $l_1$ norm-based version sees lines 9–10 of Algorithm 1 swapped with

$$\boldsymbol{v} \leftarrow \frac{\partial \boldsymbol{x}_{l+1}^{(m)}}{\left\| \boldsymbol{y}_l^{(m)} - \mu\left(\boldsymbol{y}_l^{(m)}\right) \right\|_1 / B} \qquad \partial \boldsymbol{y}_l^{(m)} \leftarrow \boldsymbol{v} - \mu(\boldsymbol{v}) - \mu\left(\boldsymbol{v} \odot \boldsymbol{x}_{l+1}^{(m)}\right) \hat{\boldsymbol{x}}_{l+1}^{(m)}, \tag{1}$$

where $B$ is the batch size. Not only does our use of $l_1$ batch normalization eliminate all squares and square roots, it also transforms one occurrence of $\boldsymbol{x}_{l+1}^{(m)}$ into its binary form, $\hat{\boldsymbol{x}}_{l+1}^{(m)}$.

**Step 2: $\boldsymbol{x}_{l+1}^{(m)}$ approximation.** Since $\partial \boldsymbol{Y}$ is quantized into our power-of-two format immediately after its calculation (Algorithm 2 line 12), we hypothesize that it should be robust to approximation. Consequently, we replace the $\boldsymbol{x}_{l+1}^{(m)}$ term remaining in (1) with the product of its signs and mean magnitude: $\hat{\boldsymbol{x}}_{l+1}^{(m)} \left\| \boldsymbol{x}_{l+1}^{(m)} \right\|_1 / B$.

Our complete batch normalization training functions are shown on lines 8–11 of Algorithm 2, which only require the storage of binary $\hat{\boldsymbol{X}}$ along with layer- and channel-wise scalars. With elements of $\boldsymbol{X}$ now binarized, we not only reduce its memory cost by $32\times$ but also save energy thanks to the corresponding memory traffic reduction.

# 6 EVALUATION

We implemented our BNN training method using Keras and TensorFlow, and experimented with the small-scale MNIST, CIFAR-10 and SVHN datasets, as well as large-scale ImageNet, using a range of network models. Our baseline for comparison was the standard BNN training method introduced by Courbariaux & Bengio (2016), and we followed those authors' practice of reporting the highest test accuracy achieved in each run. Energy consumption results were obtained using the inference energy estimator from QKeras (Coelho Jr. et al., 2020), which we extended to also estimate the energy consumption of training. This tool assumes the use of an in-order processor fabricated on a 45 nm process and a cacheless memory hierarchy, as modeled by Horowitz (2014), resulting in high-level, platform-agnostic energy estimates useful for relative comparison. Note that we did not tune hyperparameters, thus it is likely that higher accuracy than we report is achievable.

For MNIST we evaluated using a five-layer MLP—henceforth simply denoted "MLP"—with 256 neurons per hidden layer, and CNV (Umuroglu et al., 2017) and BinaryNet (Courbariaux & Bengio, 2016) for both CIFAR-10 and SVHN. We used three popular BNN optimizers: Adam (Kingma & Ba, 2015), stochastic gradient descent (SGD) with momentum and Bop (Helwegen et al., 2019). While all three function reliably with our training scheme, we used Adam by default due to its outstanding stability in performance. Experimental setup minutiae can be found in Appendix B.1.

Our choice of quantization targets primarily rested on the intuition that BNNs should be more robust to approximation in backward propagation than their higher-precision counterparts. To illustrate that this is indeed the case, we compared our method's loss when applied to BNNs vs `float32` networks with identical topologies and hyperparameters. Generally, per Table 3, significantly higher accuracy degradation was observed for the non-binary networks, as expected. While our proposed BNN training method does exhibit limited accuracy degradation—a geomean drop of 1.21 percentage points (pp) for these examples—this comes in return for simultaneous geomean memory and energy savings of $3.66\times$ and $3.09\times$, respectively, as shown in Table 4. It is also interesting to note that the training cost reductions achievable for a given dataset depend on the model chosen to classify it, as can be seen across Tables 3 and 4. This observation is largely orthogonal to our work: by applying our approach to the training of a more appropriately chosen model, one can obtain the advantages of both optimized network selection and training, effectively benefiting twice.

Table 3: Test accuracy of non-binary and BNNs using standard and proposed training approaches for various models and datasets optimized with Adam. Results for our training approach applied to the former are included for reference only; we do not advocate for its use with non-binary networks.

| Model | Dataset | Top-1 test accuracy | | | | | | |
|---|---|---|---|---|---|---|---|---|
| | | Standard training | | | Reference training | | **Proposed training** | |
| | | NN (%)[1] | BNN (%) | $\Delta$ (pp) | NN (%)[1] | $\Delta$ (pp)[2] | **BNN (%)** | $\Delta$ **(pp)**[3] |
| MLP | MNIST | 98.22 | 98.24 | 0.02 | 89.98 | −8.24 | **96.83** | **−1.41** |
| CNV | CIFAR-10 | 86.37 | 82.67 | −3.70 | 69.88 | −16.49 | **82.31** | **−0.36** |
| CNV | SVHN | 97.30 | 96.37 | −0.93 | 79.44 | −17.86 | **94.22** | **−2.15** |
| BinaryNet | CIFAR-10 | 88.20 | 89.81 | 1.61 | 76.56 | −11.64 | **88.36** | **−1.45** |
| BinaryNet | SVHN | 96.54 | 97.40 | 0.86 | 85.71 | −10.83 | **95.78** | **−1.62** |

[1] Non-binary neural network.
[2] Baseline: non-binary network with standard training.
[3] Baseline: BNN with standard training.

Table 4: Memory footprint and per-batch energy consumption of the standard and our proposed training schemes for various models using the Adam optimizer.

| Model | Memory | | | Energy/batch | | |
|---|---|---|---|---|---|---|
| | Standard (MiB) | **Proposed (MiB)** | **Saving** ($\times$) | Standard (mJ) | **Proposed (mJ)** | **Saving** ($\times$) |
| MLP | 7.40 | **2.56** | **2.89** | 2.40 | **0.97** | **2.48** |
| CNV | 128.33 | **27.13** | **4.73** | 144.24 | **52.61** | **2.74** |
| BinaryNet | 425.31 | **118.21** | **3.60** | 855.41 | **196.26** | **4.36** |

In order to explore the impacts of the various facets of our scheme, we applied them sequentially while training BinaryNet to classify CIFAR-10 with multiple optimizers. As shown in Table 5, choices of data types, optimizer and batch normalization implementation lead to clear tradeoffs against performance and resource costs. Major memory savings are attributable to the use of `float16` variables and through the use of our $l_1$ norm-based batch normalization. The bulk of our scheme's energy savings come from the power-of-two representation of $\partial Y$, which eliminates floating-point operations from lines 13–14 of Algorithm 2. We also evaluated the quantization of $\partial Y$ into five-bit layer-wise block floating-point format, denoted "`int5`" in Table 5 since the individual elements are fixed-point values. With this encoding, significantly higher accuracy loss was observed than when $\partial Y$ was quantized into the proposed, equally sized power-of-two format, confirming that representation of this variable's range is more important than its precision.

Figure 2 shows the memory footprint savings from our proposed BNN training method for different optimizers and batch sizes, again for BinaryNet with the CIFAR-10 dataset. Across all of these, we achieved a geomean reduction of 4.86×. Also observable from Figure 2 is that, for all three optimizers, movement from the standard to our proposed BNN training allows the batch size used to increase by 10×, facilitating faster completion, without a material increase in memory consumption. With respect to energy, we saw an estimated geomean 4.49× reduction, split into contributions attributable to arithmetic operations and memory traffic by 18.27× and 1.71×. Figure 2 also shows that test accuracy does not drop significantly due to our approximations. With Adam, there were small drops (geomean 0.87 pp), while with SGD and Bop we actually saw modest improvements.

We trained ResNetE-18, a mixed-precision model with most convolutional layers binarized (Bethge et al., 2019), to classify ImageNet. ResNetE-18 represents an exemplary instance within a broad class of ImageNet-capable networks, and we believe that similar results should be achievable for models with which it shares architectural features. Setup specifics can be found in Appendix B.2.

We show the performance of this network and dataset when applying each of our proposed approximations in turn, as well as with the combination we found to work best, in Table 6. Since the Tensor Processing Units we used here natively support `bfloat16` rather than `float16`, we switched to

Table 5: Accuracy, memory and energy impacts of moving from standard to our proposed data representations. We include block floating-point $\partial \mathbf{X}$ to illustrate the importance of dynamic range over precision for its representation. For these experiments, BinaryNet was trained to classify CIFAR-10.

| Optimizer | Data type | | Batch norm. | Top-1 test accuracy | | Memory saving $(\times)$[1] | Energy saving $(\times)$[1] |
|---|---|---|---|---|---|---|---|
| | $\partial \mathbf{W}$ | $\partial \mathbf{Y}$ | | % | $\Delta$ (pp)[1] | | |
| Adam | float32 | float32 | $l_2$ | 88.74 | – | – | – |
| | float16 | float16 | $l_2$ | 88.71 | −0.03 | 2.00 | 1.09 |
| | bool | float16 | $l_2$ | 87.93 | −0.81 | 2.27 | 1.10 |
| | bool | int5[2] | $l_2$ | 81.12 | −7.62 | 2.50 | 4.32 |
| | bool | po2_5 | $l_2$ | 89.47 | 0.73 | 2.50 | 4.01 |
| | bool | po2_5 | $l_1$ | 87.64 | −1.10 | 2.50 | 4.01 |
| | bool | po2_5 | Proposed | **88.59** | **−0.15** | **3.60** | **4.36** |
| SGD with momentum | float32 | float32 | $l_2$ | 88.52 | – | – | – |
| | float16 | float16 | $l_2$ | 88.54 | 0.02 | 2.00 | 1.09 |
| | bool | float16 | $l_2$ | 87.35 | −1.17 | 2.31 | 1.10 |
| | bool | int5 | $l_2$ | 81.89 | −6.63 | 2.59 | 4.40 |
| | bool | po2_5 | $l_2$ | 89.08 | 0.56 | 2.59 | 4.06 |
| | bool | po2_5 | $l_1$ | 88.69 | 0.17 | 2.59 | 4.06 |
| | bool | po2_5 | Proposed | **87.45** | **−1.07** | **4.07** | **4.45** |
| Bop | float32 | float32 | $l_2$ | 91.38 | – | – | – |
| | float16 | float16 | $l_2$ | 91.36 | −0.02 | 2.00 | 1.09 |
| | bool | float16 | $l_2$ | 90.54 | −0.84 | 2.37 | 1.10 |
| | bool | int5 | $l_2$ | 40.55 | −50.83 | 2.72 | 4.48 |
| | bool | po2_5 | $l_2$ | 89.34 | −2.04 | 2.72 | 4.11 |
| | bool | po2_5 | $l_1$ | 87.81 | −3.57 | 2.72 | 4.11 |
| | bool | po2_5 | Proposed | **86.28** | **−5.10** | **4.92** | **4.53** |

[1] Baseline: float32 $\partial \mathbf{W}$ and $\partial \mathbf{X}$ with standard ($l_2$) batch normalization.
[2] 5-bit fixed-point elements of layer-wise block floating-point format.

the former for these experiments. Where bfloat16 variables were used, these were employed across all layers; the remaining approximations were applied only to binary layers. Despite increasing the precision of our power-of-two quantized $\partial \mathbf{Y}$ by moving from $k = 5$ to 8, this scheme unfortunately induced significant accuracy degradation, suggesting incompatibility with large-scale datasets. Consequently, we disapplied it for our final experiment, which saw our remaining three approximations deliver memory and energy reductions of $3.12\times$ and $1.17\times$ in return for a 2.25 pp drop in test accuracy. While these savings are smaller than those of our small-scale experiments, we note that ResNetE-18's first convolutional layer is both its largest and is non-binary, thus its activation storage dwarfs that of the remaining layers. We also remark that, while $\sim$2 pp accuracy drops may not be acceptable for some application deployments, sizable training resource reductions are otherwise possible. The effects of binarized $\partial \mathbf{W}$ are insignificant since ImageNet's large images result in proportionally small weight memory occupancy. Nevertheless, this proof of concept demonstrates the feasibility of large-scale neural network training on the edge.

## 7  CONCLUSION

In this paper, we introduced the first training scheme tailored specifically to BNNs. Moving first to 16-bit floating-point representations, we selectively and opportunistically approximated beyond this based on careful analysis of the standard training algorithm presented by Courbariaux & Bengio. With a comprehensive evaluation conducted across multiple models, datasets, optimizers and batch sizes, we showed the generality of our approach and reported significant memory and energy reductions vs the prior art, challenging the notion that the resource constraints of edge platforms present insurmountable barriers to on-device learning. In the future, we will explore the potential of our training approximations in the custom hardware setting, within which we expect there to be vast energy-saving potential through the design of tailor-made arithmetic operators.

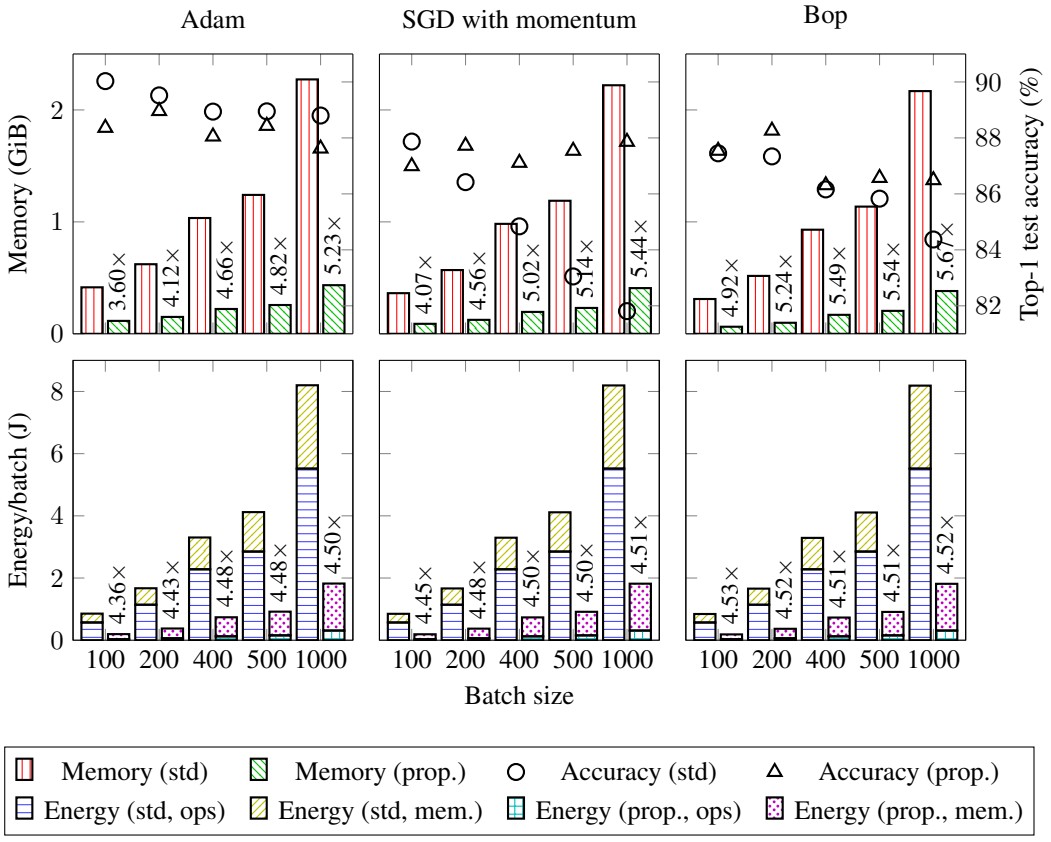

Figure 2: Batch size vs training memory footprint, achieved test accuracy and per-batch training energy consumption for BinaryNet with CIFAR-10. The upper plots show memory and accuracy results for the standard and our proposed training flows. In the lower plots, total energy is split into compute- and memory-related components. Annotations show reductions vs the standard approach.

Table 6: Test accuracy, memory footprint and per-batch energy consumption of the standard and our proposed training schemes for ResNetE-18 classifying ImageNet with Adam used for optimization.

| Approximations | Top-1 test accuracy | | Memory | | Energy/batch | |
|---|---|---|---|---|---|---|
| | % | $\Delta$ (pp)[1] | GiB | Saving ($\times$)[1] | J | Saving ($\times$)[1] |
| None | 58.57 | – | 57.84 | – | 185.08 | – |
| All-`bfloat16` | 58.55 | 0.02 | 29.32 | 1.97 | 162.41 | 1.14 |
| `bool` $\partial W$ only | 57.30 | −1.27 | 57.80 | 1.00 | 185.08 | 1.00 |
| `po2_8` $\partial Y$ only | 29.56 | −29.01 | 57.84 | 1.00 | 116.06 | 1.59 |
| $l_1$ batch norm. only | 57.34 | −1.23 | 57.84 | 1.00 | 185.08 | 1.00 |
| Proposed batch norm. only | 57.25 | −1.32 | 35.59 | 1.63 | 176.87 | 1.05 |
| **Final combination**[2] | **56.32** | **−2.25** | **18.54** | **3.12** | **158.44** | **1.17** |

[1] Baseline: approximation-free training.
[2] `bool` $\partial W$ and `bfloat16` remaining variables with proposed batch normalization.

ACKNOWLEDGMENTS

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

# A  CONVERGENCE RATE ANALYSIS

## A.1  THEORETICAL SUPPORT

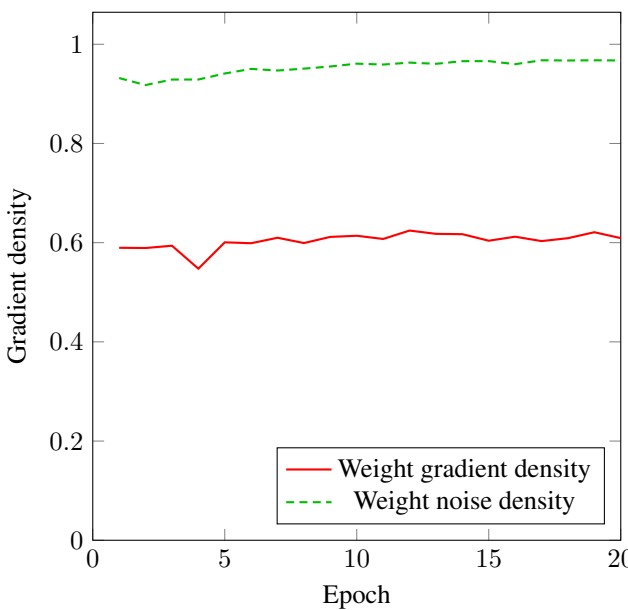

Figure 3: Weight density of the sixth convolutional layer of BinaryNet trained with `bool` weight and `po2_5` activation gradients using Adam and the CIFAR-10 dataset.

Bernstein et al. (2018) proved that training non-binary networks with binary weight gradients may result in similar convergence rates to those of unquantized training if *weight gradient density* $\phi\left([\mu(\partial\boldsymbol{W}_1),\cdots,\mu(\partial\boldsymbol{W}_S)]^{\mathrm{T}}\right)$ and *weight noise density* $\phi\left([\sigma(\partial\boldsymbol{W}_1),\cdots,\sigma(\partial\boldsymbol{W}_S)]^{\mathrm{T}}\right)$ remain within an order of magnitude throughout a training run. Here, $S$ is the training step size and $\phi(\bullet) = \frac{\|\bullet\|_1^2}{N\|\bullet\|_2^2}$ denotes the density function of an $N$-element vector.

We repeated Bernstein et al.'s evaluation with our proposed gradient quantization applied during BinaryNet training with the CIFAR-10 dataset using Adam and hyperparameters as detailed in Appendix B.1. The results of this experiment can be found in Figure 3. We chose to show the densities of BinaryNet's sixth convolutional layer since this is the largest layer in the network. Each batch of inputs was trained using quantized gradients $\partial\hat{\boldsymbol{W}}$ and $\partial\tilde{\boldsymbol{Y}}$. The trained network was then evaluated using the same training data to obtain the `float32` (unquantized) $\partial\boldsymbol{W}$ used to plot the data shown in Figure 3. We found that the weight gradient density ranged from 0.55–0.62, and weight noise density 0.92–0.97, therefore concluding that our quantization method may result in similar convergence rates to the unquantized baseline.

It should be noted that Bernstein et al.'s derivations assumed the use of smooth objective functions. Although the forward propagation of BNNs is not smooth due to binarization, their training functions still assume smoothness due to the use of straight-through estimation.

## A.2  EMPIRICAL SUPPORT

Figures 4, 5, 6 and 7 contain the training accuracy curves of all experiments conducted for this work. The curves of the standard and our proposed training methods are broadly similar, supporting the conclusion from Appendix A.1 that our proposals do not induce significant convergence rate change.

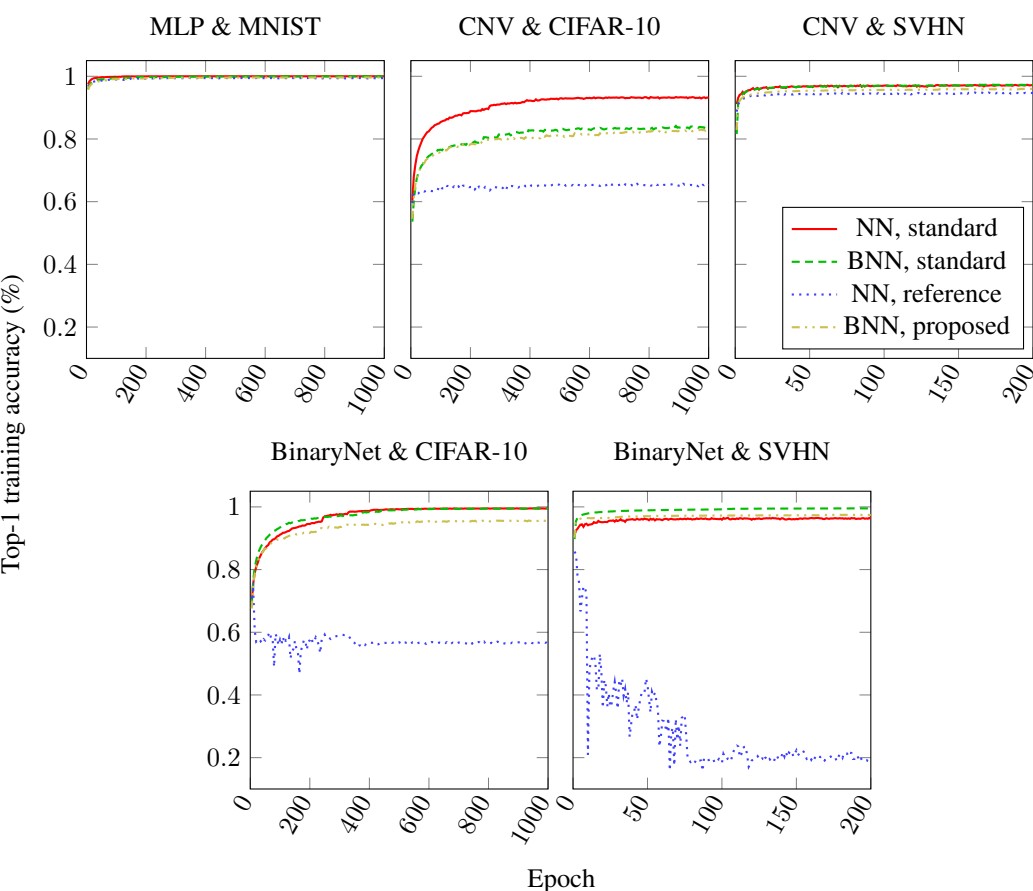

Figure 4: Achieved training accuracy over time for experiments reported in Table 3.

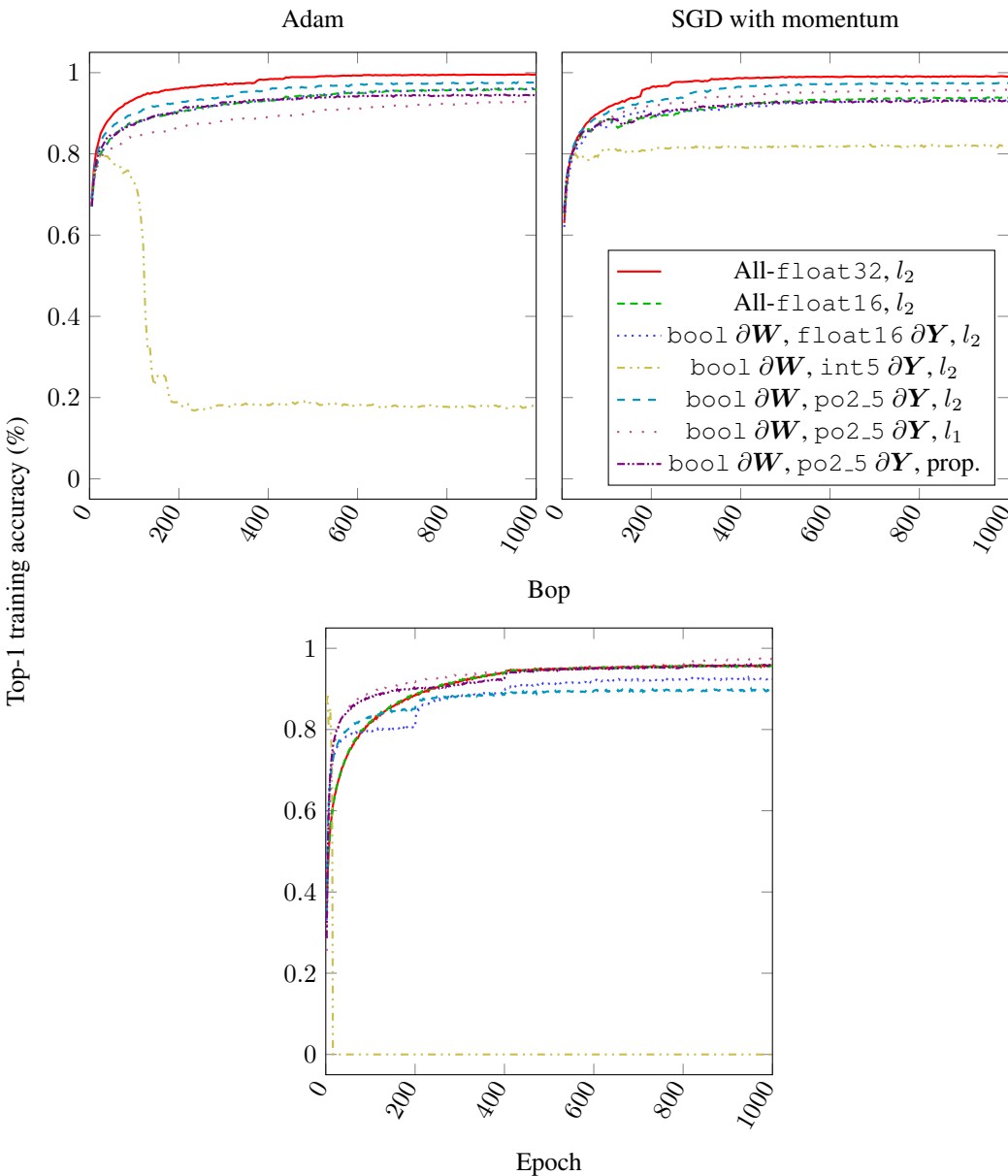

Figure 5: Achieved training accuracy over time for experiments reported in Table 5.

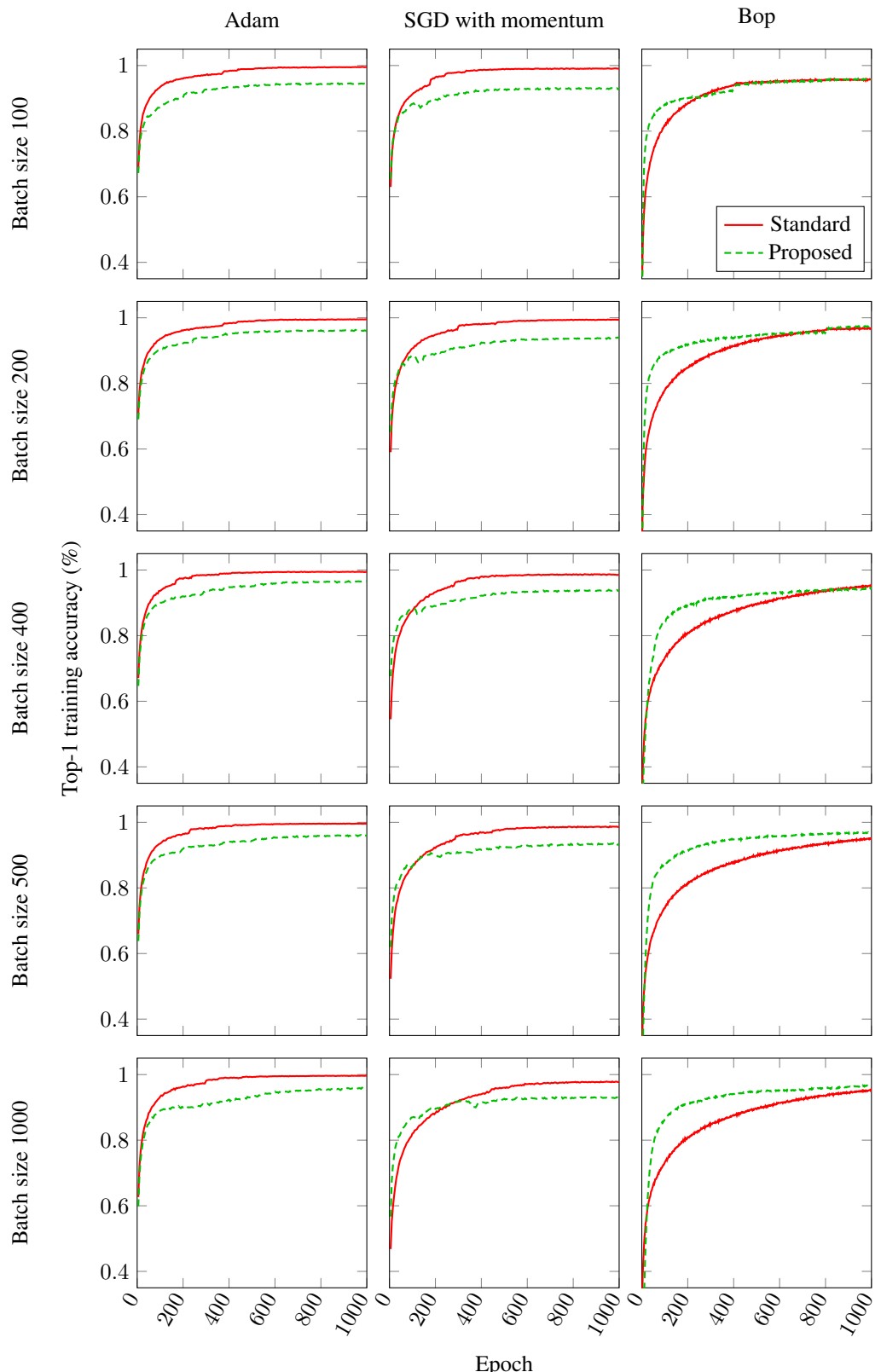

Figure 6: Achieved training accuracy over time for experiments reported in Figure 2.

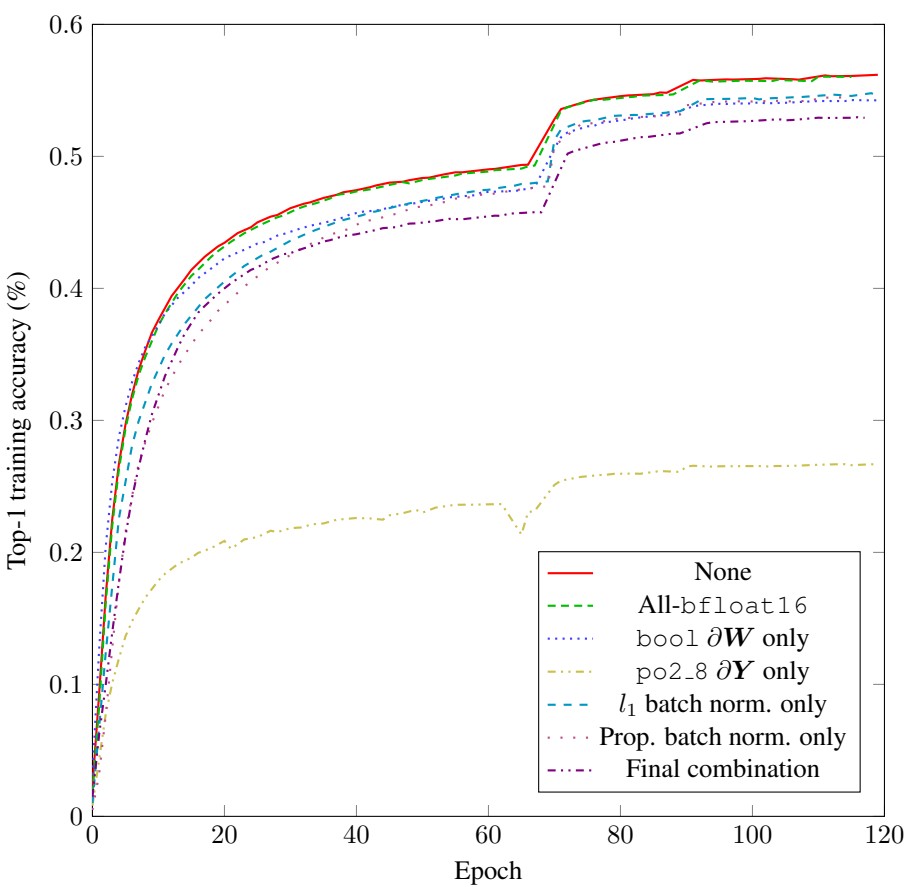

Figure 7: Achieved training accuracy over time for experiments reported in Table 6.

## B  EXPERIMENTAL SETUP

### B.1  SMALL-SCALE DATASETS

We used the development-based learning rate scheduling approach proposed by Wilson et al. (2017) with an initial learning rate $\eta$ of 0.001 for all optimizers except for SGD with momentum, for which we used 0.1. We used batch size $B = 100$ for all except for Bop, for which we used $B = 50$ as recommended by Helwegen et al. (2019). MNIST and CIFAR-10 were trained for 1000 epochs; SVHN for 200.

### B.2  IMAGENET

Finding development-based learning rate scheduling to not work well with ResNetE-18, we resorted to the fixed decay schedule described by Bethge et al. (2019). $\eta$ began at 0.001 and decayed by a factor of 10 at epochs 70, 90 and 110. We trained for 120 epochs with $B = 4096$.

