# OpenReview forum: "Enabling Binary Neural Network Training on the Edge"
_ICLR.cc/2021/Conference — Reject_

### Official Review · AnonReviewer4 · 2020-10-26
**I think this paper is very interesting and if some minor comments are addressed, it should be accepted for presentation at ICLR21**

**Rating:** 8
**Confidence:** 5

**Review:**

I think this a very good contribution to ICLR given the topic and the quality of the submission (originality, contribution to the state of the art, experimental evidence, e

 Some of the strong points of the submission are summarized as follows, along with some points for clarification

1.	Enabling training on any embedded device (SoC, FPGA, micro-controller) is one of the holy grails for edge AI and IoT. As the authors mention, this also intersects with other domains such as federating learning privacy by design systems. The authors provide and ample motivations of the importance of this work, and some of the applications edge AI might enable, as well as the current challenges.
2.	The state of the art (despite the previous comment) contextualizes the subject matter in a succinct but comprehensive manner. Although there are certain aspects that could be improved, such as including a table outlining in a clearer manner the contributions of the authors in this context.
3.	The comparison with the traditional training method is clear. However, I would like to know if the authors have made an ablation study to assess whether or not the use of batch normalization would have an effect on the accuracy of the proposed models. Do you provide experimental evidence of the lack of degradation due to the use of l1 batch normalization? These two aspects are not mentioned in the next nor provided in the supplementary sections.
4.	The experimental design is good, showing a careful analysis to validate the proposal and several ablation studies to assess the memory footprint reductions and its effects on the training of various models
5.	The foundations for the method are presented in great detail in a formalized manner and provides sufficient elements (i.e. experiments) to assess the validity of the proposed approach.

---

> ### Author Response · Authors · 2020-11-17
> **Response to Reviewer 4**
>
> >Although there are certain aspects that could be improved, such as including a table outlining in a clearer manner the contributions of the authors in this context.
>
> Thank you for this suggestion. We now compare more directly against prior low-cost training works targeting non-binary networks in the newly added Table 1.
>
> >However, I would like to know if the authors have made an ablation study to assess whether or not the use of batch normalization would have an effect on the accuracy of the proposed models. Do you provide experimental evidence of the lack of degradation due to the use of l1 batch normalization?
>
> Authors including Sari et al. (2019) have reported that, without batch normalization, BNNs suffer from gradient explosion. We confirmed this observation in our early-stage experiments with BinaryNet, as we now state explicitly in Section 3. While we included results showing accuracy change as a result of our $l_1$-based batch normalization in Tables 4 and 5 (now 5 and 6), we now further include results for $l_1$ batch normalization in the style proposed by Wu et al. (2018b), i.e. Section 5.2 step 1 only, in Tables 4 and 5 (now 5 and 6).

---

### Official Review · AnonReviewer3 · 2020-10-28
**This work proposes a low-cost BNN training scheme to reduce memory consumption and improve energy efficiency. Compared to the traditional BNN training algorithm, the proposed scheme achieves a significant reduction in memory footprint and energy consumption on multiple datasets.**

**Rating:** 5
**Confidence:** 3

**Review:**

I agree with the key contributions listed in the paper, especially the binarization of weight gradients and activations. The paper is well written and clearly articulates a contribution to the literature. The proposed BNN training scheme can have a significant practical impact. The experimental evidence is provided for several standard image classification tasks. Most of the related works are cited. The paper does not contain a theory part, but wherever possible, equations are provided to illustrate how the method works.

Concerns:
This paper includes a detailed empirical evaluation of the proposed BNN training scheme. The major concern is that the proposed low-cost BNN training scheme can cause a nontrivial accuracy degradation (2.25%) as shown in Table 5. The tradeoff between accuracy and memory footprint/energy consumption is not carefully evaluated. For example, a smaller network model with fewer parameters and activations can be trained using the baseline BNN training scheme to reduce memory and energy consumption. Besides, the baseline networks (BinaryNet and ResNet) used for comparison are out-of-date.  Many recent works [1-3] propose new BNN architectures, which improve the accuracy of BNNs significantly. It is useful to justify the effectiveness of the proposed scheme for these SOTA BNN architectures.

To evaluate the energy consumption of the traditional and the proposed BNN training scheme, the authors assume that the two training schemes have the same convergence rate. Although the traditional BNN training scheme consumes higher power during training, it might take fewer epochs to reach the same accuracy. Therefore, the saving in energy consumption can be lower than the reported numbers. In addition, the estimated energy consumption of BNN training obtained from using QKeras is very rough, which makes the improvement in energy consumption less convincing. The authors can mainly improve the paper's strength by prototyping the proposed BNN training scheme on an embedded CPU and measuring real-world performance and power.

Reasons for score: In general, I like the idea of enabling low-cost BNN training by identifying unnecessary high-precision data. However, the improvement numbers presented in the paper need better justification. I would consider raising my score if the authors could address the aforementioned concerns.

[1] Bi-Real Net: Enhancing the Performance of 1-bit CNNs With Improved Representational Capability and Advanced Training Algorithm

[2] ProxyBNN: Learning Binarized Neural Networks via Proxy Matrices

[3] ReActNet: Towards Precise Binary Neural Network with Generalized Activation Functions

---

> ### Author Response · Authors · 2020-11-17
> **Response to Reviewer 3**
>
> >The major concern is that the proposed low-cost BNN training scheme can cause a nontrivial accuracy degradation (2.25\%) as shown in Table 5.
>
> As also shown in Table 5 (now 6), this degradation comes in return for memory and energy consumption reductions of 3.12$\times$ and 1.17$\times$, respectively. Whether or not such an accuracy degradation is acceptable will be application-dependent, as we now make explicit in Section 6. In Tables 3 and 5 (now 4 and 6), we found broadly similar degradation across models spanning a range of sizes, thereby demonstrating the generality of our approach.
>
> >The tradeoff between accuracy and memory footprint/energy consumption is not carefully evaluated. For example, a smaller network model with fewer parameters and activations can be trained using the baseline BNN training scheme to reduce memory and energy consumption.
>
> These tradeoffs can be found in Tables 4 and 5 (now 5 and 6). While training cost reductions are possible through the selection of different network models, this observation is largely orthogonal to our work: by applying our approach to the training of a smaller model, one can obtain the advantages of both optimized network selection and training, effectively benefiting twice. In Tables 3 and 5 (now 4 and 6), we showed significant savings across models spanning a range of sizes, thereby demonstrating the generality of our approach. We have rephrased and expanded the relevant discussion in Section 6 to more clearly emphasize these points.
>
> >Besides, the baseline networks (BinaryNet and ResNet) used for comparison are out-of-date. Many recent works [1-3] propose new BNN architectures, which improve the accuracy of BNNs significantly. It is useful to justify the effectiveness of the proposed scheme for these SOTA BNN architectures.
>
> Since the works the reviewer highlights share architectural features with networks for which we obtained positive results -- particularly ResNetE-18's skip connections -- we see no fundamental reason why our approach would not be favorable with these models as well. Moreover, ResNetE-18 was published in 2019, and is therefore more current than Bi-Real Net (2018). We now cite the works highlighted by the reviewer in Section 2, and have added text to Section 6 to explain that ResNetE-18 is representative of a broader class of modern network models.
>
> >To evaluate the energy consumption of the traditional and the proposed BNN training scheme, the authors assume that the two training schemes have the same convergence rate. Although the traditional BNN training scheme consumes higher power during training, it might take fewer epochs to reach the same accuracy. Therefore, the saving in energy consumption can be lower than the reported numbers.
>
> We included a subset of our experiments' training curves in Appendix A.2 to show that our method does not result in convergence rate deterioration vs the baseline. To further support this claim, we now include the training curves for all of our experiments in Appendix A.2, and have highlighted the lack of induced convergence rate deterioration in the abstract and Section 1.
>
> >In addition, the estimated energy consumption of BNN training obtained from using QKeras is very rough, which makes the improvement in energy consumption less convincing. The authors can mainly improve the paper's strength by prototyping the proposed BNN training scheme on an embedded CPU and measuring real-world performance and power.
>
> We chose to use QEnergy since it provides platform-agnostic energy estimates. While we acknowledge that a platform-specific implementation would have allowed us to obtain more accurate energy consumption figures than reported in the paper, we believe that our QEnergy-derived estimates are more useful from a high-level perspective in quantifying relative energy changes in a platform-agnostic manner. We have rephrased our description of the energy estimator in Section 6 to emphasize this point.

---

### Official Review · AnonReviewer2 · 2020-10-28

**Rating:** 6
**Confidence:** 4

**Review:**

#### Comments
Summary:

The authors proposed a low-cost binary neural network training strategy exhibiting sizable memory footprint reductions and energy savings. The methods include binarizing weight gradients, modifying the forward and backward batch normalization operations and using power-of-two activation gradients and reduced-precision floating-point data. The experimental results show some improvement memory footprint reductions and energy savings vs standard approach.


Strength:
-- The authors carried out a relatively sufficient experimental analysis, and evaluated across multiple models, data sets, optimizers and batch sizes
-- Storage and energy consumption based on hardware models increases the completeness and credibility of conclusions.
-- The experiment results seem that the proposed method achieves good performance in memory footprint reductions and energy savings.

Weakness:
-- The proposed method seems like a combination of existing technical methods.
-- There is a lack of comparison with other low-cost binary neural network training works.


Comments:
(1)	There is a difference in the memory consumption in Table1 and in section 5.1 (1.67 MiB or 1.41 MiB ?). The authors may check this.
(2)	As for the perspective of overall design, it’s better to emphasize the trade-offs between the importance of different variables to the overall training and the choose of the data type.
(3)	It’s better to add some comparisons with other low-cost binary neural network training works.

---

> ### Author Response · Authors · 2020-11-17
> **Response to Reviewer 2**
>
> >The proposed method seems like a combination of existing technical methods.
>
> We would like to emphasize that this work does not represent simply a combination of approximation techniques but, as detailed in Sections 1, 4 and 5, a number of novel insights into BNNs' robustness to particular quantization schemes resulting in careful selection of these methods. In order to more clearly highlight our work's novelty, we now compare our use of approximation vs previous non-binary neural network training works in the newly added Table 1.
>
> >There is a lack of comparison with other low-cost binary neural network training works.
>
> While there are no prior works specifically addressing the training costs of BNNs, we compared against works targeting training cost reductions of non-binary networks in Tables 4 and 5 (now 5 and 6). In Table 5 (now 6), for example, "bool $\partial\boldsymbol{W}$ only" is equivalent to SignSGD (Bernstein et al., 2018). In order to further highlight the benefits of our BNN-specific approach, we have added results for $l_1$ batch normalization in the style proposed by Wu et al. (2018b), i.e. Section 5.2 step 1 only, to Tables 4 and 5 (now 5 and 6). Furthermore, we now compare our use of approximation vs previous non-binary neural network training works in the newly added Table 1.
>
> >(1) There is a difference in the memory consumption in Table1 and in section 5.1 (1.67 MiB or 1.41 MiB ?). The authors may check this.
>
> Thank you for pointing out this discrepancy. We have amended the value in Section 5.1 to the correct 1.67 MiB.
>
> >(2) As for the perspective of overall design, it’s better to emphasize the trade-offs between the importance of different variables to the overall training and the choose of the data type.
>
> These tradeoffs can be found in Tables 4 and 5 (now 5 and 6). We have rephrased the text related to these tables in Section 6 to better emphasize the tradeoffs.
>
> >(3) It’s better to add some comparisons with other low-cost binary neural network training works.
>
> Please find our response to this comment above.

---

### Official Review · AnonReviewer1 · 2020-10-28
**Official Blind Review #1**

**Rating:** 5
**Confidence:** 3

**Review:**

This paper proposes a novel method to make the training Binary Neural Networks with low-memory and low-energy by modifying the backpropagation and forward process. To this end, the paper binarized weight gradients, change batch normalization layer for removing full-precision the inputs, and utilize quantization to accelerate the whole training BNNs procedure.

The paper targeted one of the problems in Binary Neural Networks and provided experiments as well as source codes to the proof of efficiency. This is the most significant contribution of the paper.

However, there are the following concerns:
- Need more experiments on the actual training time between two BNNs which are trained with quantized and full-precision gradient weight.
- Comparison with similar works in recent BNNs.
- What does it mean B variables in Algorithm 2 of lines: 6, 9, and 10, and how does it influence the performance of training?
- In figure 2, the legends should be put in the figure (not in the caption). It is better to follow.
- It would be nice to have the training time and accuracy in the large-scale dataset of ImageNet.

In conclusion, the paper addresses the novel idea for the training improvement of Binary Neural Networks in low-memory and low-energy. However, there are many concerns aforementioned.

---

> ### Author Response · Authors · 2020-11-17
> **Response to Reviewer 1**
>
> >Need more experiments on the actual training time between two BNNs which are trained with quantized and full-precision gradient weight.
>
> We did not include training time results since the purpose of this work is to reduce the resource costs of BNN training, not to accelerate it. We did, however, include a subset of our experiments' training curves in Appendix A.2 to show that our method does not result in convergence rate deterioration vs the baseline. To further support this claim, we now include the training curves for all of our experiments in Appendix A.2, and have highlighted the lack of induced convergence rate deterioration in the abstract and Section 1.
>
> >Comparison with similar works in recent BNNs.
>
> While there are no prior works specifically addressing the training costs of BNNs, we compared against works targeting training cost reductions of non-binary networks in Tables 4 and 5 (now 5 and 6). In Table 5 (now 6), for example, "bool $\partial\boldsymbol{W}$ only" is equivalent to SignSGD. In order to further highlight the benefits of our BNN-specific approach, we have added results for $l_1$ batch normalization in the style proposed by Wu et al. (2018b), i.e. Section 5.2 step 1 only, to Tables 4 and 5 (now 5 and 6). Furthermore, we now compare our use of approximation vs previous non-binary neural network training works in the newly added Table 1.
>
> > What does it mean B variables in Algorithm 2 of lines: 6, 9, and 10, and how does it influence the performance of training?
>
> $B$ is the batch size, which we have clarified in Section 3. We exemplified the impact on accuracy, memory and energy of varying $B$ in Figure 2.
>
> >In figure 2, the legends should be put in the figure (not in the caption). It is better to follow.
>
> Thank you for raising this point, which we have now addressed for all of our plots.
>
> >It would be nice to have the training time and accuracy in the large-scale dataset of ImageNet.
>
> ImageNet accuracy results can be found in Table 5 (now 6). While we did not include ImageNet training time results since the purpose of this work is to reduce the resource costs of training, not to accelerate it, we did include a subset of our experiments' training curves in Appendix A.2 to show that our method does not result in convergence rate deterioration vs the baseline. To further support this claim, we now include the training curves for all of our experiments in Appendix A.2, and have highlighted the lack of induced convergence rate deterioration in the abstract and Section 1.

---

### Author Response · Authors · 2020-11-17
**General Responses**

We thank the reviewers for their positive assessment of our work and helpful suggestions for improvement.
Please find our responses to specific questions directly after each review.

Reviewers questioned the lack of comparison with other low-cost BNN training works. We would like to kindly point out that there are no prior works specifically addressing the training costs of BNNs. As a result, we compared against prior works targeting training cost reduction for non-binary networks in Tables 4 and 5 (now 5 and 6). We now include additional results within the same tables to further highlight the benefits of our BNN-specific approach.

We have also added training curves to Appendix A.2, now covering all experiments reported in the paper, to more robustly highlight that our proposals do not result in convergence rate deterioration vs Courbariaux \& Bengio's standard approach.

---

### Decision · Program_Chairs · 2021-01-07
**Final Decision**

**Decision:**

Reject

**Comment:**

After reading the paper, reviews and authors’ feedback. The meta-reviewer agrees that this paper addresses an important topic. However, as the reviewers pointed out. The paper mainly builds the technique on simulated setting, and it is unclear how the method will translate to real world speedups. Past work(e.g. [1]) has also shown that many cases there could be a huge gap when the solution is not built carefully.

The paper would benefit from a prototype to demonstrate the applicability of the approach. This paper is therefore rejected.

Thank you for submitting the paper to ICLR.

[1] Riptide: Fast End-to-End Binarized Neural Networks

---

> ### Author Response · Authors · 2021-01-18
> **Response to the Program Chairs**
>
> We thank the reviewers and program chairs for their feedback on our work. We would like to highlight that our objective was to save *memory* during network training -- the most common resource limitation on edge devices -- and that we did not set out to achieve or make any claims regarding *speedup*. Algorithms 1 and 2, Table 2 and our supplementary materials capture our calculations of memory usage. We believe these results are sufficient to demonstrate the memory reduction claimed, and that therefore building a prototype would not add further scientific value.